# Patients and healthcare workers' preferences for using smart lockers in accessing and dispensing chronic disease medication in Nigeria: Findings from a descriptive cross-sectional study

Ibrahim Bola Gobir[1]*, Samson Agboola[2], Havilah Onyinyechi Nnadozie[2], Helen Adamu[2], Fatimah Ohunene Sanni[2], Aisha Adamu[2], Azeez Akanbi Bello[2], Angmun Suzzy Otubu[3], Deus Bazira[1], Piring'ar Mercy Niyang[1]

1 Center for Global Health Practice and Impact, Georgetown University, Washington, DC, United States of America, 2 Georgetown Global Health Nigeria, Federal Capital Territory, Abuja, Nigeria, 3 Savannah Health System Innovation Limited, Federal Capital Territory, Abuja, Nigeria

* ibg7@georgetown.edu

## Abstract

The use of smart locker technology has been beneficial for patients with chronic diseases who require regular medication and face challenges accessing healthcare facilities due to distance, time, or mobility issues. This study aimed to assess preferences for utilizing Smart Lockers in accessing and dispensing chronic disease medication among healthcare workers (HCWs) and patients in Nigeria. A descriptive cross-sectional survey was conducted between November 8th and December 4th, 2021, across secondary healthcare facilities in five states of Adamawa, Akwa Ibom, Cross River, Benue, and Niger. Among 1,133 participants included in the analysis, 405 were HCWs and 728 were patients with chronic illnesses. Descriptive statistics, including frequencies and percentages, were used to summarize the data, while chi-square tests were employed to assess significant differences between healthcare workers (HCWs) and patients. Results indicated a strong preference among both HCWs and patients for one-on-one counseling as the preferred method for orientating patients on using Smart Lockers, with 53.8% of HCWs and 58.1% of patients expressing this preference (p = 0.25). Additionally, there was a shared preference for hospitals or clinics as secure locations for Smart Lockers, with 68.9% of HCWs and 71.6% of patients preferring this option (p < 0.05). The majority of participants favored receiving notification of drug delivery via phone call, with 49.1% of HCWs and 48.8% of patients expressing this preference (p = 0.63). There was a significant difference in preferences for access hours, the majority (HCWs: 65.4% and patients: 52.6%) favored 24-hour access (p < 0.05). Participants identified patients with HIV within the age range of 18–40 as the most suitable population to benefit from using Smart Lockers for medication dispensing. These findings offer insights into healthcare policies aimed at enhancing medication access and adherence among patients with chronic diseases in Nigeria. The development of models for using

**Data Availability Statement:** Data set supporting the findings of this study are available within the paper and its Supporting information files.

**Funding:** -IBG -GU-HRP-502 - Georgetown ICF_Smart Locker_IBG_12142021 -Georgetown University Medical Center Dean of Research -https://cghpi.georgetown.edu/ -No, the funders had no role in the study design, data collection and analysis, decision to publish or preparation of the manuscript.

**Competing interests:** The authors have declared that no competing interests exist.

smart lockers to dispense chronic disease medications to chronically ill persons in Nigeria and other populations is recommended.

## Background

Chronic disease is a sickness that lasts a long time, advances slowly, is caused by genetics, the environment, or poor lifestyle choices, and requires continuous medical care [1]. Chronic diseases were responsible for more than 28 million deaths worldwide in 1990. By 2008, this number had climbed to 36 million, and 39 million in 2016 [2]. Chronic diseases remain the leading cause of death globally [3, 4]. Evidence suggests that chronic disease patients often experience multimorbidity because chronic diseases rarely occur in isolation. The burden of managing the treatment of chronic illnesses includes changes in lifestyle, and behaviours, regular visits to the healthcare professional or clinic, and managing medications [5].

In Nigeria, medication dispensing for chronic diseases traditionally occurs within healthcare facilities, reflecting the historical development of the healthcare system. Patients typically obtain medications directly from hospitals or clinics during visits where healthcare professionals oversee the process. The impact of chronic diseases is worse in developing countries like Nigeria due to limitations in technological advancement for prompt diagnosis and treatment [6]. So therefore, there's increasing recognition of the need for innovative approaches like smart lockers to improve accessibility and adherence in managing chronic diseases [7].

In the health care context, smart lockers are automated machines that enable patients to access medications without delay rather than wait long hours in the queue at the health facility or pharmacy. The lockers can be placed at locations like shopping malls, retail stores, cinemas, etc., where users can easily access them [8]. Smart lockers and pharmaceutical dispensing devices are newer technologically driven processes that have been tested on a small scale. To decongest the clinics and reduce waiting time, Neo Hutiri developed the Pelebox smart locker, which allows patients access to the automated drug dispensing machine using an automated unique PIN-generating model for patient use only. This one-time unique PIN allows patients to retrieve medication from the machine These automated machines have been tested in Johannesburg, South Africa [9].

Smart lockers have remarkably reduced long waiting hours at health facilities, reduced the financial burden on patients and facilities, and increased adherence to medication among Persons Living with HIV (PLHIV) and chronic disease patients in South Africa. Introducing a similar system in Nigeria will reduce the burden of health services provided to the already overwhelmed healthcare workforce and largely improve the healthcare system [10]. This is because no previous studies have identified, documented, or evaluated using smart lockers in Nigeria for dispensing chronic disease medication. Therefore, this study aimed to assess the patients' and healthcare workers' preferences for using Smart Lockers in accessing and dispensing chronic disease medications in Nigeria [11, 12].

## Method

### Study design and setting

This study employed a cross-sectional study to assess preferences for using Smart Lockers in accessing and dispensing chronic disease medication in Nigeria among patients and healthcare workers (HCWs). The study took place between November 8th and December 4th, 2021,

across secondary healthcare facilities in five states: Adamawa, Akwa Ibom, Cross River, Benue, and Niger.

These healthcare facilities were selected based on their specialization in providing care for patients with chronic diseases such as Diabetes, hypertension, HIV, cancer, and Tuberculosis, among others. Identification of these facilities was facilitated through collaboration with Implementing Partners supporting clinical care initiatives in these states.

## Study population

The target populations for this study were:

1. **Health Workers:** This group includes healthcare professionals responsible for administering medical care and services to individuals diagnosed with chronic diseases.

2. **Persons Living with Chronic Illnesses:** This group consists of individuals who have been diagnosed with one or more chronic diseases such as HIV, diabetes, hypertension, chronic kidney disease, Cancer, TB, etc.

## Inclusion criteria

1. People aged 18 years or above who have been diagnosed with a chronic disease requiring medication refills such as HIV, diabetes, hypertension, chronic kidney disease, Cancer, TB, etc. and receive care in the facilities listed in the S1 Appendix.

2. Healthcare workers responsible for providing care to individuals with chronic illnesses at the facilities listed in the S1 Appendix.

## Sample size determination

**Healthcare workers (HCWs).**   According to a research paper on the health workforce estimated between 2016 to 2030 to understand if Nigeria will have enough workforce [13], the estimated number of HCWs was 621,205. This was used as a proxy for HCWs providing care to chronic illness patients. This was inputted into the Raosoft sample size calculator at a 5% margin of error, 95% confidence interval and 50% response distribution to yield a minimum sample size of 384.

**Patients of persons living with chronic illnesses.**   According to a research paper on patterns of chronic illnesses conducted in Nigeria in 2020 [14], the percentage of chronic diseases in Nigeria is about 64.9% of the population. This was used as a proxy to estimate the population prevalence of chronic diseases given that the population of Nigeria is about 200 million [15]. This was entered into the Raosoft sample size calculator at 4% margin of error, 95% confidence interval and 50% response distribution to yield a minimum sample size of 601.

**Selection of participants.**   The selection of participants involved identifying healthcare workers providing chronic disease care and patients with chronic illnesses through the selected healthcare facilities in the respective states. Stratified random sampling was used to ensure representation, and participants were stratified based on gender (50% male and 50% female) to enhance diversity and inclusivity.

## Data collection

Data collection involved the administration of a semi-structured questionnaire to both healthcare workers and patients living with chronic diseases. The questionnaire comprised 37 items designed to gather information on sociodemographic characteristics, accessibility preferences,

methods of engagement, operability, and eligible populations that would be most suited to use Smart Lockers for accessing and dispensing chronic disease medication.

Participants were briefed about the survey's objectives and were required to give their consent before completing the questionnaire. The questionnaire included a section on sociodemographic characteristics, covering gender, age, marital status, and highest educational qualification. No personally identifiable data was collected from the participants.

## Data analysis

The data analysis comprised descriptive statistics, including frequencies and percentages, to summarize the demographic characteristics such as gender, age, marital status, and highest educational qualification and responses obtained from both healthcare workers (HCWs) and patients regarding their preferences for accessing and dispensing chronic disease medication using Smart Locker. Furthermore, a chi-square test was conducted to evaluate significant differences between the preferences of HCWs and patients. The significance level was set at $p < 0.05$, with a 95% confidence interval utilized as the threshold for identifying statistically significant findings. All analyses were performed at a 5% significance level using statistical software, Stata version 15.0 and Microsoft Excel (2016), to ensure accuracy and reliability in the interpretation of the data.

## Ethical statement

**Informed consent.**   Participants were provided with detailed information regarding the study objectives, eligibility criteria, data privacy measures, and researchers' disclaimers on the entry page of the survey. Informed consent was obtained from all participants before proceeding with the survey. Consent was indicated by ticking the "I agree" checkbox on the survey, indicating the participant's voluntary agreement to participate in the study. Entries from participants who did not meet the inclusion criteria were not processed for data analysis, ensuring that only eligible participants' data were included.

**Confidentiality.**   To uphold participant confidentiality, all entries were recorded anonymously. Personally identifiable information was not collected from participants during the survey. Privacy of the subjects' information was strictly maintained throughout the study duration.

**Risks and benefits.**   Participants were informed that there were no adverse effects associated with participating in the study that would compromise their rights or welfare. Additionally, there were no direct benefits offered to participants for their involvement in the study.

## Results

A total of 1,180 survey entries were received, and 47 (4.0%) responses were excluded from data analysis because they did not meet the inclusion criteria. Of the 1,133 responses included in the analysis, 405 were received from HCWs, and 728 were from persons living with chronic illnesses in Nigeria.

Table 1 and (Fig 1) show the demographic characteristics of the participants with chronic diseases and healthcare workers in 5 states in Nigeria. Most of the respondents in the patient survey were female (55.9%), while a higher percentage of healthcare workers (HCWs) were male (51.4%). In terms of age distribution, 67.9% of healthcare workers (HCWs) were between the age group of 18–35 years., whereas 50.1% of patients were aged between 36 and 60 years 48.9% of the healthcare workers (HCWs) were single, and 57.6% of patients were married. Moreover, most participants had Post-Secondary Education (86.9%) among healthcare

**Table 1. Demographics of the participants of patients with chronic diseases and health care workers in 5 states in Nigeria.**

| Variables | HCWs, n = 405 | Patients, n = 728 |
|---|---|---|
| **Gender** | | |
| Male | 208 (51.4%) | 321 (44.1%) |
| Female | 197 (48.6%) | 407 (55.9%) |
| **Age** | | |
| 18–35 | 275 (67.9%) | 338 (46.4%) |
| 36–60 | 125 (30.9%) | 365 (50.1%) |
| > 60 | 5 (1.2%) | 25 (3.4%) |
| **Marital Status** | | |
| Single | 198 (48.9%) | 187 (25.7%) |
| Married | 197 (48.6%) | 419 (57.6%) |
| Separated | 10 (2.5%) | 122 (16.8%) |
| **Highest Educational Qualification** | | |
| No formal Education | 2 (0.5%) | 97 (13.3%) |
| Primary Education | 7 (1.7%) | 135 (18.5%) |
| Secondary Education | 44 (10.9%) | 295 (40.5%) |
| Post-Secondary Education | 352 (86.9%) | 201 (27.6%) |

workers (HCWs), while a higher percentage had completed Secondary Education (40.5%) among patients.

Among healthcare workers (HCWs), (Table 2) and (Fig 2) indicate that pharmacists or other pharmacy staff were the most preferred personnel to load drugs into the smart lockers, with a response rate of 64.7%. This preference was significantly higher compared to the Monitoring and Evaluation (M&E) staff, who had the lowest response rate at 1.5%. The next most preferred personnel were those specifically employed for this purpose, with HIV case managers having a response rate of 16.3% and other specific staff at 14.1%. In contrast, nurses and other professions had lower response rates at 1.7%.

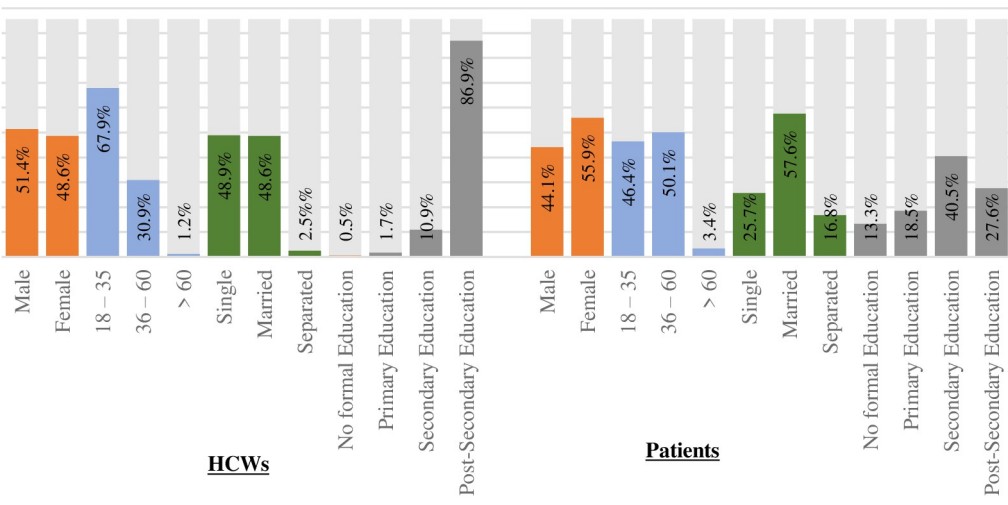

**Fig 1. Demographic characteristics of the participants.**

**Table 2.  The most suitable personnel to load the drugs into the smart lockers in 5 states in Nigeria.**

| Personnel | HCWs, n = 405 |
|---|---|
| HIV Case Manager | 57 (14.1%) |
| Monitoring and Evaluation (M&E)staff | 6 (1.5%) |
| Nurse | 7 (1.7%) |
| Pharmacist or other pharmacy staff | 262 (64.7%) |
| Specific staff employed for this sole purpose | 66 (16.3%) |
| Others | 7 (1.7%) |

Others: Trained personnel and staff

The preference for one-on-one counselling in (Fig 3) was identified as the best method for orientating patients on using Smart Lockers for accessing and dispensing chronic disease medication. This was significantly high among both healthcare workers (HCWs) at 53.8% and patients at 58.1%. This shared preference suggests a strong inclination towards utilizing Smart

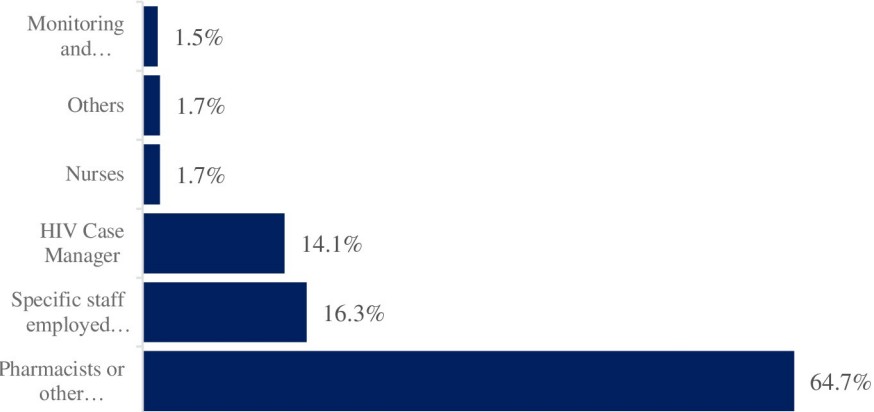

**Fig 2. The most suitable personnel to load the drugs into the smart lockers.**

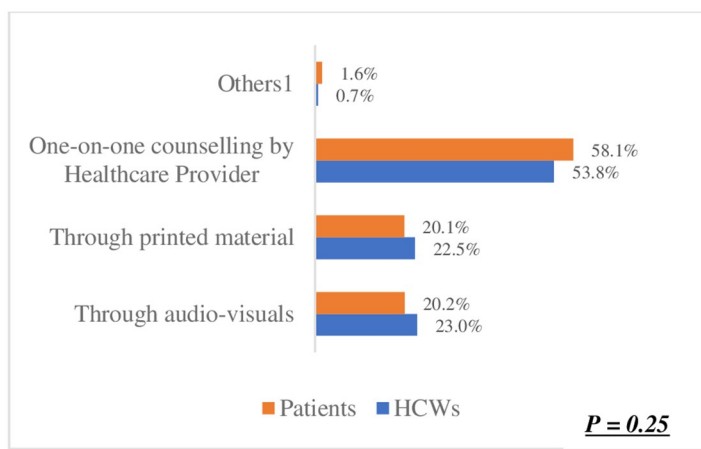

**Fig 3. Preferred orientation method.**

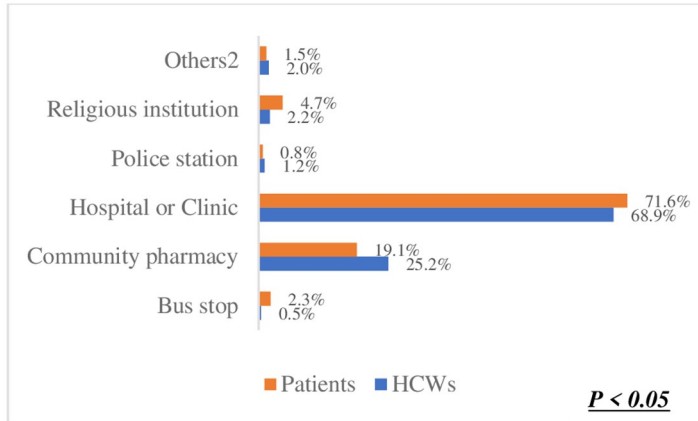

**Fig 4. The best location for accessibility.**

Lockers for medication dispensing. Moreover, there was no significant difference observed between HCWs and patients regarding this preference (p = 0.25). Regarding the preferred location for smart lockers in (Fig 4), a hospital or clinic emerged as the top choice among participants. Patients exhibited a higher preference for this option (71.6%) compared to HCWs (68.9%). In (Fig 5), the difference was statistically significant (p < 0.05),in terms of notification methods for drug delivery. The majority of both HCWs (49.1%) and patients (48.8%) favoured receiving a phone call. Notably, there was no significant difference between HCWs and patients in this regard (p = 0.63). However, in (Fig 6) a difference was observed concerning access hours for Smart Lockers. While the majority (HCWs: 65.4% and patients: 52.6%) favoured 24-hour access for drug pick-up, there was a significant difference between the two populations (p < 0.05). In (Fig 7), participants identified patients with HIV as the most

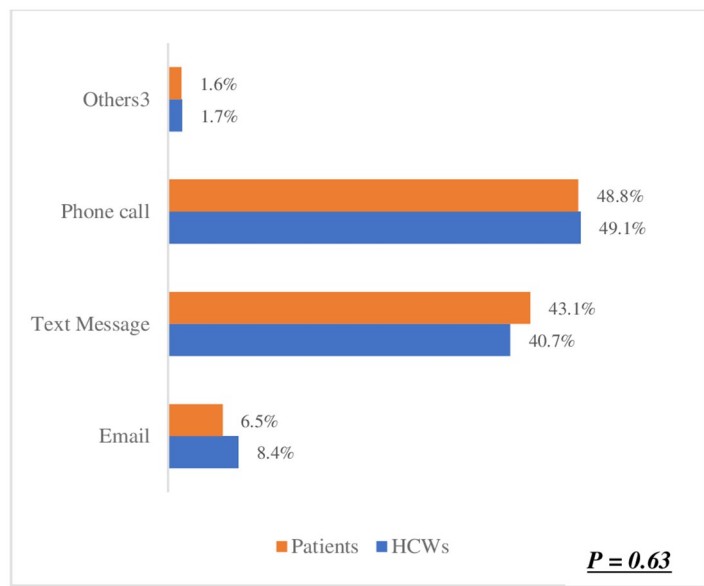

**Fig 5. Preferred notification for drug pick-up.**

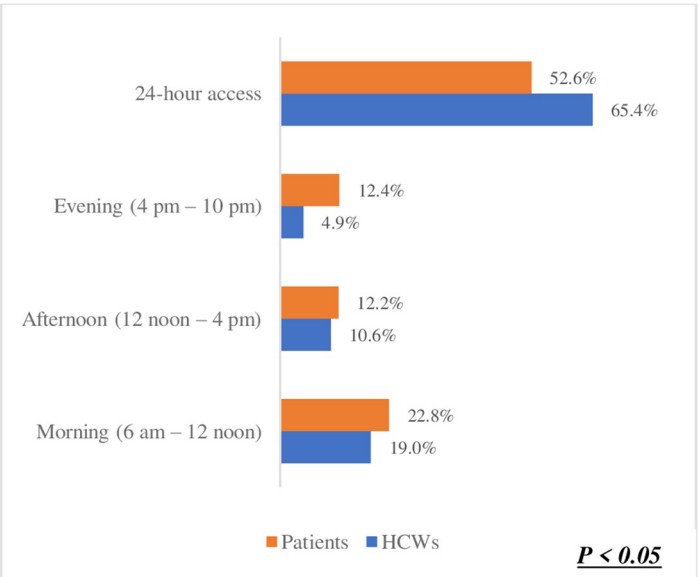

**Fig 6. Preferred access hours.**

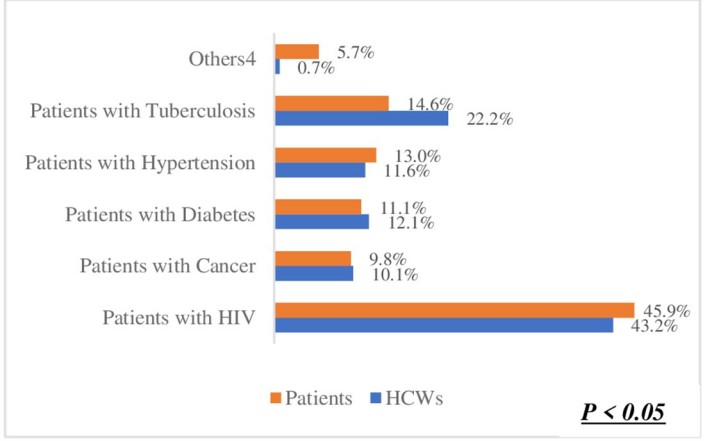

**Fig 7. Patients who might benefit.**

suitable population to benefit from using smart lockers(HCWs: 43.2%, Patients: 45.9%) and finaly in (Fig 8), participants within the age range of 18–40 were identified to have the highest number of prefarred usage for smart lockers (HCWs: 81.0%, Patients: 70.5%), (Table 3).

## Discussion

The study explored patients' and healthcare workers' preferences for using Smart Lockers in accessing and dispensing chronic disease medications in Nigeria. Before now, there has been a need for research aimed at evaluating the efficacy of a system like smart lockers in the Nigerian healthcare space. Part of the justification for our study stems from the absence of prior investigations into the utilization of smart lockers for the distribution of chronic disease medication

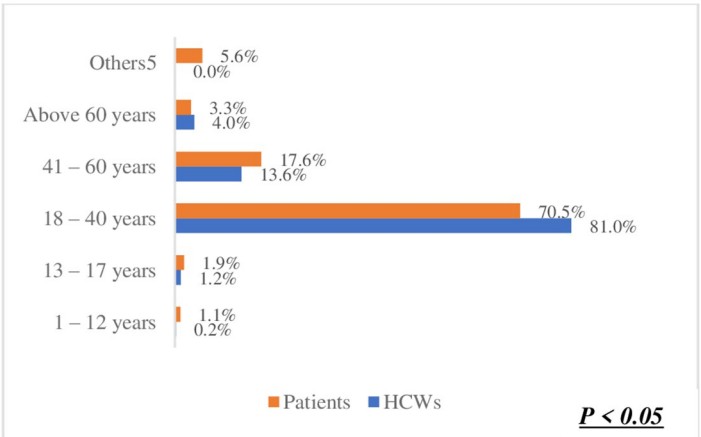

**Fig 8. Age groups to use smart lockers.**

within Nigeria. According to our survey findings, smart lockers can be strategically placed close to a secure clinic or hospital, allowing patients in the surrounding area 24-hour access to pick up their prescription drugs. Medication can be administered by calling the patients. Patients, or end users, need to be trained on how to use the smart lockers; one-on-one counselling from their HCW was identified as the best course of action. According to Angell and others [16], the quality of healthcare systems in low-middle-income countries like Nigeria is poor. This contributes to the increase in the burden of illnesses and global health costs. There is an increasing focus on strengthening health systems by investing in improving the quality of health services and creating equitable access to high-quality, people-centred care in building trust in the health system. To achieve these goals, the perspectives, and preferences of patients and HCWs must be considered [17].

The preference from patients and HCWs regarding the populations best suited to use the Smart Lockers for chronic disease dispensing indicates that persons living with HIV between 18 and 40 years are the most suitable populations to use the Smart Lockers. This is likely due to the prevalence of HIV in Nigeria, with the highest HIV prevalence rates recorded among women between the ages of 25 and 34 [18]. This may also be because most respondents in the survey were between the ages of 18 and 40. Additionally, decentralized care, such as community-based drug delivery and ART refills from community pharmacies, is popular in managing diseases such as HIV and Tuberculosis in Nigeria, unlike other chronic diseases [19]. This would indicate a lesser resistance to adopting smart lockers as another model for decentralized care. However, the incidence of other chronic diseases, such as diabetes, hypertension, and stroke, is higher in older populations aged 36–60 and above 60 years [20, 21]. There is increasing advocacy for improving person-centred care for older adults with chronic illnesses by transferring care from healthcare facilities to home and community-based models [22].

While most respondents advocate placing the lockers within a health institution such as a hospital, clinic, or community pharmacy, very few were open to using other locations such as bus stops, religious institutions, and police stations. This may be due to concerns relating to safety, privacy, and stigmatization [23]. Literature suggests that the rates of stigmatization differ based on the health condition, with persons suffering from HIV, Tuberculosis, and mental health conditions being among the most severely affected by stigmatization. These factors are often linked to a lack of awareness, beliefs, and institutionalized practices within healthcare

**Table 3. Preferences for the use of smart lockers in dispensing chronic disease medications in 5 states in Nigeria.**

| Variables | HCWs, n = 405 | Patients, n = 728 | p-value |
|---|---|---|---|
| **How might patient orientation be conducted on the use of the smart lockers?** | | | |
| Through audio-visuals | 93 (23.0%) | 147 (20.2%) | p = 0.25 |
| Through printed material | 91 (22.5%) | 146 (20.1%) | |
| One-on-one counselling by Healthcare Provider | 218 (53.8%) | 423 (58.1%) | |
| Others[1] | 3 (0.7%) | 12 (1.6%) | |
| **What would be the best location for placing the smart lockers for access to patients?** | | | |
| Bus stop | 2 (0.5%) | 17 (2.3%) | p < 0.05 |
| Community pharmacy | 102 (25.2%) | 139 (19.1%) | |
| Hospital or Clinic | 279 (68.9%) | 521 (71.6%) | |
| Police station | 5 (1.2%) | 6 (0.8%) | |
| Religious institution | 9 (2.2%) | 34 (4.7%) | |
| Others[2] | 8 (2.0%) | 11 (1.5%) | |
| **How best might patients receive notification of drug dispensing?** | | | |
| Email | 34 (8.4%) | 47 (6.5%) | p = 0.63 |
| Text Message | 165 (40.7%) | 314 (43.1%) | |
| Phone call | 199 (49.1%) | 355 (48.8%) | |
| Others[3] | 7 (1.7%) | 12 (1.6%) | |
| **What operating time would you recommend for patients to access the smart lockers?** | | | |
| Morning (6 am– 12 noon) | 77 (19.0%) | 166 (22.8%) | p < 0.05 |
| Afternoon (12 noon– 4 pm) | 43 (10.6%) | 89 (12.2%) | |
| Evening (4 pm– 10 pm) | 20 (4.9%) | 90 (12.4%) | |
| 24-hour access | 265 (65.4%) | 383 (52.6%) | |
| **What patients might benefit from using the smart lockers to collect their medication?** | | | |
| Patients with HIV | 175 (43.2%) | 334 (45.9%) | p < 0.05 |
| Patients with Cancer | 41 (10.1%) | 71 (9.8%) | |
| Patients with Diabetes | 49 (12.1%) | 81 (11.1%) | |
| Patients with Hypertension | 47 (11.6%) | 95 (13.0%) | |
| Patients with Tuberculosis | 90 (22.2%) | 106 (14.6%) | |
| Others[4] | 3 (0.7%) | 41 (5.7%) | |
| **For which of these age groups can the smart lockers be used in dispensing medication?** | | | |
| 1–12 years | 1 (0.2%) | 8 (1.1%) | p < 0.05 |
| 13–17 years | 5 (1.2%) | 14 (1.9%) | |
| 18–40 years | 328 (81.0%) | 513 (70.5%) | |
| 41–60 years | 55 (13.6%) | 128 (17.6%) | |
| Above 60 years | 16 (4.0%) | 24 (3.3%) | |
| Others[5] | 0 (0.0%) | 41 (5.6%) | |

Others[1]: Unspecified.

Others[2]: Any convenient location, Bank, Post office.

Others[3]: Religious announcements, cluster collectors, disseminating information to other patients, face-to-face, home visits.

Others[4]: Asthmatic, Surgical cases, Malaria, Hepatitis with stroke

Others[5]: Unspecified

facilities and communities [24, 25]. Decentralizing care and bringing the lockers to non-health-related institutions further expose patients to stigmatization and lack of privacy risks.

The patients and HCWs reported that they preferred smart lockers in a hospital or clinic. The sentiments behind this preference may be similar to those reported by Peng and others

[26], who found that patients living with chronic diseases preferred to access care through primary healthcare (PHC) services, provided that the distance to the service provider was short and that they had access to capable HCWs. The use of designated areas in the community close to the patients, such as religious institutions or retail pharmacies for drug pick-ups for chronic non-communicable diseases, is reported to limit the patient's need for physician consultation to issues such as an error in prescription, delay in delivery and adverse drug reactions [27].

Results of our study also indicated that one-on-one counselling would be the most effective method of orientating patients using smart lockers. This may be linked to factors such as the literacy level of patients and their comprehension of health-related information disseminated through mediums such as audio-visual tools and printed materials. Most patients in our study reported having completed a secondary school education (40.5%). Patient literacy has been suggested to influence individuals' health-seeking attitudes, impacting their ability to read, understand and act upon health-related information [28]. Additionally, many patients rely heavily on the opinions of their HCWs and are more likely to adopt treatment measures that a trusted physician suggests. This is especially applicable in the case of chronically ill persons who require ongoing care and have developed a strong relationship with their HCWs over time [29]. Having one-on-one contact with a Healthcare provider may overcome the barriers to understanding information about using smart lockers and increase their likelihood of adopting smart lockers for collecting their medication. A study by Donelan and others [30] indicates that healthcare services should be made available to patients at any time of the day, considering the technological advancement the world is experiencing. Telemedicine affords a patient access to the physician without time restrictions and not necessarily having to go into the hospital [31]. This points to the agreement found in this study from HCW and patients' responses to access time to smart lockers. Most patients and HCWs have the same line of thought that smart lockers for drug dispensing should be placed at the hospitals or clinics and made operational for 24 hours for easy access.

Furthermore, the usage of questionnaires may have resulted in self-selection bias, as respondents may have an essentially distinctive characteristic that affected their decision to prefer or not to prefer the use of smart lockers.

Due to the characteristics of the hospitals chosen and the fact that the HIV program provides the most extensive organized chronic care programs in those locations, the study has some limitations that may limit the generalizability of the findings to other chronic diseases.

In our study, most respondents disclosed that a phone call would be the best method for notifying patients when their drugs are dispensed. The use of smartphones has been reported to enhance patient drug adherence. Short Message Service (SMS) and synchronized data transmission applications enable healthcare providers to send reminders and keep up with patients' drug adherence [32]. However, the ownership of mobile phones and smartphones in Nigeria is limited. Limited network access in certain areas may also hinder the quick and seamless transmission of electronic mail and SMS [33]. This may be considered the reason why most of the respondents prefer receiving a phone call notification when their drugs are dispensed. This would be easier to track and ensure that the patients are adequately informed when their prescriptions are dispensed.

Proxy estimates were used to determine sample sizes and assess chronic disease prevalence, potentially introducing bias. Collaboration with specific healthcare facilities and Implementing Partners may lead to selection bias, limiting generalizability. The cross-sectional design prevents establishing causality in the study. Self-reported data may be subject to social desirability or recall bias, impacting response accuracy. The study's focus on specific states may limit its applicability to broader populations in Nigeria.

## Conclusion

Our study provides valuable insights into the preferences for utilizing Smart Lockers in accessing and dispensing chronic disease medication among healthcare workers (HCWs) and patients in Nigeria. The strong preference for one-on-one counselling as the preferred method for orienting patients on using Smart Lockers underscores the importance of personalized education and support in implementing new healthcare technologies. Additionally, the shared preference for hospitals or clinics as secure locations for Smart Lockers highlights the significance of infrastructure and accessibility in medication management initiatives. The majority of participants revealed that receiving notification of drug delivery via phone call, differences in preferences for access hours suggest the need for tailored approaches to accommodate varying schedules and preferences among HCWs and patients. Notably, patients with HIV within the age range of 18–40 were identified as the most suitable population to benefit from Smart Lockers usage, indicating potential areas for targeted intervention and implementation strategies.

Overall, these findings contribute to the growing body of literature on innovative approaches to medication access and adherence in chronic disease management. By understanding and addressing the preferences and needs of both HCWs and patients, healthcare policymakers and providers can optimize the implementation of Smart Lockers technology to enhance medication accessibility and improve health outcomes among individuals living with chronic diseases in Nigeria.

## Supporting information

**S1 File. Feasibility and acceptability of "smart lockers" for dispensing chronic disease medication in Nigeria.**
(PDF)

**S2 File.**
(XLSX)

**S3 File. STROBE statement—Checklist of items that should be included in reports of** *cross-sectional studies.*
(DOCX)

**S4 File. PLOS ONE clinical studies checklist.**
(DOCX)

**S5 File. Inclusivity in global research.**
(DOCX)

**S1 Appendix. Distribution of states and facilities.**
(DOCX)

## Acknowledgments

We would like to thank all those who contributed in one way or another to the collection and analysis of this data.

## Author Contributions

**Conceptualization:** Ibrahim Bola Gobir, Angmun Suzzy Otubu, Deus Bazira, Piring'ar Mercy Niyang.

**Data curation:** Samson Agboola, Havilah Onyinyechi Nnadozie, Azeez Akanbi Bello, Angmun Suzzy Otubu.

**Formal analysis:** Samson Agboola, Azeez Akanbi Bello.

**Funding acquisition:** Ibrahim Bola Gobir, Deus Bazira, Piring'ar Mercy Niyang.

**Investigation:** Samson Agboola, Havilah Onyinyechi Nnadozie, Helen Adamu.

**Methodology:** Samson Agboola, Havilah Onyinyechi Nnadozie.

**Project administration:** Havilah Onyinyechi Nnadozie, Helen Adamu, Fatimah Ohunene Sanni, Piring'ar Mercy Niyang.

**Resources:** Ibrahim Bola Gobir, Angmun Suzzy Otubu, Deus Bazira.

**Supervision:** Ibrahim Bola Gobir, Helen Adamu, Piring'ar Mercy Niyang.

**Visualization:** Piring'ar Mercy Niyang.

**Writing – original draft:** Ibrahim Bola Gobir, Havilah Onyinyechi Nnadozie.

**Writing – review & editing:** Helen Adamu, Fatimah Ohunene Sanni, Aisha Adamu.

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
