## [Decision Letter · Decision Letter 0]

2 Jan 2024

PONE-D-23-39195Patients and Healthcare Workers' Preferences for Using Smart Lockers in Dispensing Chronic Disease Medication in NigeriaPLOS ONE

Dear Dr. Gobir,

Thank you for submitting your manuscript to PLOS ONE. After careful consideration, we feel that it has merit but does not fully meet PLOS ONE’s publication criteria as it currently stands. Therefore, we invite you to submit a revised version of the manuscript that addresses the points raised during the review process.

We look forward to receiving your revised manuscript.

Kind regards,

Mohammed Shuaib

Academic Editor

PLOS ONE

Journal Requirements:

2.Please include a complete copy of PLOS’ questionnaire on inclusivity in global research in your revised manuscript. Our policy for research in this area aims to improve transparency in the reporting of research performed outside of researchers’ own country or community. The policy applies to researchers who have travelled to a different country to conduct research, research with Indigenous populations or their lands, and research on cultural artefacts. The questionnaire can also be requested at the journal’s discretion for any other submissions, even if these conditions are not met.  Please find more information on the policy and a link to download a blank copy of the questionnaire here: https://journals.plos.org/plosone/s/best-practices-in-research-reporting. Please upload a completed version of your questionnaire as Supporting Information when you resubmit your manuscript.

3. Please expand the acronym “GUMC” (as indicated in your financial disclosure) so that it states the name of your funders in full.

Reviewers' comments:

Reviewer's Responses to Questions

**Comments to the Author**

1. Is the manuscript technically sound, and do the data support the conclusions?

Reviewer #1: Yes

Reviewer #2: Partly

Reviewer #3: Yes

2. Has the statistical analysis been performed appropriately and rigorously? 

Reviewer #1: Yes

Reviewer #2: Yes

Reviewer #3: Yes

3. Have the authors made all data underlying the findings in their manuscript fully available?

Reviewer #1: Yes

Reviewer #2: Yes

Reviewer #3: Yes

4. Is the manuscript presented in an intelligible fashion and written in standard English?

Reviewer #1: Yes

Reviewer #2: Yes

Reviewer #3: Yes

5. Review Comments to the Author

Reviewer #1: This is an important work on dispensing medication for chronic disease through the use of smart lockers. The authors did a good job of putting in context and justifying the need for such kind of service in Nigeria or similar contexts in sub-Saharan Africa. The authors aimed to describe preferences for using smart lockers in dispensing medication for chronic disease patients and potential caregivers of these patients.

Comments and Issues

1. It would been better if this study was enlarged into a feasibility study and used mixed methods approaches. That would enrich the quantitative data. This is just a lamenting comment and nothing can be done about it.

2. Make a single background section:

- Remove the subdivisions such as the "problem statement", "justification", and "aim and objectives" i.e merge all these portions

- Revise the text to remove redundancies. For example, current lines 131 to 133 come back again in lines 135 and 136 (despite the slight rewrite) and it comes again as aim and objectives. There are a lot of these types of repetitions.

3. About the sample size determination:

- The two references 9 and 23 do not provide any information on how the sample size was calculated. Citation 9 is actually for a single hospital.

- Also, what is the expected point estimate wished to be estimated with a 5% margin error?

- Lines 184 to 185 are for data collection. Move them.

4. Line 66 in the current background, is written: "Nigerians have a 20% chance of dying untimely from chronic diseases". This is a wrong interpretation of the "probability of dying between the age of 30 and 70 years from NCD". Please see the WHO definition of premature mortality from NCD.

5. In the data analysis subsection

- Line 196. Do not write STATA it is Stata. See the Stata documentation. Ok?

- Please clarify the line 195.

6. Results section:

- Table 1 is it possible to add for the HCW their cadres (eg: physicians, nurses etc)? For the patients, it would be good if we had their type of NCD.

- Table 4 is basically a repetition of table 2. Only p-values were added. I do not think we need these p-values.

Reviewer #2: Title:

Patients and Healthcare Workers' Preferences for Using Smart Lockers in Dispensing

Chronic Disease Medication in Nigeria

Overall comments

I congratulate the authors on the manuscript that assessed patients and health worker preferences for using smart lockers. More client centered models of service delivery are needed now more than ever as people living with HIV age with the disease and present with chronic diseases. In this regard, this paper addresses a very important topic as more integrated models of care for chronic diseases are sought. The paper is well written. Below are a few comments for clarification.

Title: would be great if the title included the study design used. The revised title could be “Patients and Healthcare Workers' Preferences for Using Smart Lockers in Dispensing Chronic Disease Medication in Nigeria—findings from a descriptive cross-sectional study” or something similar.

Abstract: Summarizes the study but it’s difficult to follow the research the way results are described. The results for HCWs and for patients need to be presented separately. It’s not clear what patients and clients were asked about. How many patients had different chronic diseases?

While the abstract mentions 6 states but only five are listed namely, Adamawa, Akwa Ibom, Cross River, Benue, and Niger. Line 30, add “individuals or participants or respondents” after 1,133. Line 31 and 32, the percentages should be replaced with absolute number or used in addition to absolute numbers. Line 32, the sentence that ends “are the most suitable population to use the smart lockers” is written like a conclusion rather than a finding. Please clarify what the sentence means or rewrite.

Does this statement “Out of the 180 responses received for” mean that the questionnaires were sent by mail? Does the statement also imply that some research respondents were included even though they were not HCWs or were not living with an NCD? The results section of the abstract needs to be restructured with findings on preferences presented separately for HCWs and for clients.

It’s not clear from reading the findings how the 1st concluding statement was reached at.

Introduction: Generally, well written but can be shortened. I find line 91-97 irrelevant or too broad and could be excluded so that smart lockers are described in the health care context. The “problem statement” and “justification” are all parts of the introduction. The subheadings are not necessary in the manuscript. Rather, the two sections should be consolidated into the introduction. Similarly, the aims and objectives can be added to the introduction usually as the last statement. E.g. This study described……... There is also no need for numbering since it’s one objective.

Methods:

The methods section is very comprehensive. As mentioned elsewhere, the multiple small headings are distractive. Several statements need to be contextualized as non-Nigerian readers may not know what secondary facilities are. So, a brief description of the levels of health facilities and referral patterns would be useful including what services are provided at primary, secondary and tertiary center.

Similarly, implementing partners should be explained as it applies to only HIV. Line 155, mention if this was a self-administered questionnaire. Clarify if HCWs and patients answered the same questionnaire. If not, what differed in the questionnaires.

Line 162, write in past tense. For patients, please clarify the age criteria. Were children excluded?

162-164: The two sentences should be combined by saying. This study included patients with chronic diseases and HCWs who manage them.

Sampling: Apart from patients with HIV, it’s not clear how patients with other chronic conditions were selected since to the best of my knowledge HIV patients receive care in different clinics from those with other conditions. Was there a deliberate attempt to select people from different clinics? Or were only people with HIV and another chronic condition selected?

Ethical statement: should just indicate that informed consent was obtained from all participants. Not sure what line 202 refers to or whether it’s needed.

Results are well written. The tables are clear, but they should be comprehensively labeled including more details, for examples Table 1: “Demographics of the participants”. The correct label should be “ Demographics of the participants pf patients with chronic diseases and HCWs in 5 states in Nigeria, YYYY). Apply this kind of naming to all tables and figures.

Line 223, it’s difficult to believe that any HCWs would be 18 years of age. Please clarify and the methods section could include what HCWs were included nurses, doctors etc. and why these were selected.

Table 2, the statement “The patients that might benefit from using the smart lockers to collect their medication” should be rewritten as “What patients that might benefit from using the smart lockers to collect their medication”. While this question is useful for HCWs, it was not right to ask people living with a chronic disease who they thought would benefit from the lockers.

Table 2 and 4 seem duplicative. Could be consolidated.

Line 280-282, what difference is being described?

Discussion:

A typical discussion should start with a description of what the study was about and what was done. Based on this the discussion should be rewritten to discuss the results rather than a general argument.

Reviewer #3: 1. More information is required in the background on traditional method of dispensing in order to bring out clearly its' challenges and support the need for an advanced technology;

2. 5 states were listed in the methods and not 6 as was stated in the preceding abstract;

3. It was not clear why these states were chosen and not others. There are 36 states in Nigeria and I will expect a spread in the choice but here we have 2 states from South West alone. This could represent bias in the study. Again, the chosen states are not the most viable socio=economically to support a new dispensing technology. The states are not the most populated such that there will be many outpatients that will stay for long hours to receive medications;

4. PLWHIV require specialized care from Consultants, which could only be obtained in tertiary health facilities in Nigferia and not secondary;

5. To qualify as a 'Nigerian' research, a state is to be selected from each of the 6 geopolitical zones of the country, else you should simply add 'some' or 'certain' States in Nigeria;

6. It was not clear whether the questionnaire is self administered or interviewer as as many in the patient group have only secondary education or below as their qualification and may not be able to comprehend the questionnaire easily;

7. Are these smart lockers manually or digitally operated?

6. PLOS authors have the option to publish the peer review history of their article (what does this mean?). If published, this will include your full peer review and any attached files.

Reviewer #1: **Yes: **Orvalho Augusto

Reviewer #2: No

Reviewer #3: No

---

## [Author Response · Author response to Decision Letter 0]

12 Feb 2024

Dear Dr Orvalho Augusto, Reviewer 2 and Reviewer 3,

We express our gratitude for your valuable feedback and the chance to submit a revised draft of our manuscript, now titled Patients and Healthcare Workers' Preferences for Using Smart Lockers in Dispensing Chronic Disease Medication in Nigeria—Findings from a Descriptive Cross-Sectional Study. The meticulous attention to detail, as well as the time and resources dedicated to offering constructive comments and reviews, is sincerely appreciated. We have diligently incorporated changes to address the majority of the suggestions provided, ensuring that we contextualize some of the comments within the paper. Your input has significantly enhanced the quality and comprehensiveness of our work.

The following are the responses to the journal requirements: 

1. The manuscript has been adjusted to reflect the PLOS One filing system requirement.

2. The PLOS questionnaire on inclusivity in global research does not apply because the research was conducted locally.

3. The acronym GUMC has been expanded accordingly. 

4. Captions have been included at the end of the document. 

Reviewer 1 (Orvalho Augusto): Thank you for your response, the subdivisions in this section have been merged. Lines 131 to 133 and lines 135 to 136 have been adjusted. 

Reviewer 1 (Orvalho Augusto): Thank you for your response, the 50% expected response point is included in the sample size determination.

Reviewer 1 (Orvalho Augusto): We are appreciative of your feedback. The statement has been corrected to reflect the WHO definition of premature mortality.

Reviewer 1 (Orvalho Augusto): Thank you for pointing this out in line 195. It has been corrected.

Reviewer 1 (Orvalho Augusto): Thank you for your response. During the study, we considered the HCW as a unit providing care to patients living with chronic diseases, therefore, we did not collect the cadre of the participating HCWs, we only sampled their opinion on the suitable personnel for loading drugs into the smart lockers. 

Table 2 and 4 have been harmonized.

Reviewer 2: Thank you for your suggestion on the topic. the title has been adjusted to reflect “Patients and Healthcare Workers' Preferences for Using Smart Lockers in Dispensing Chronic Disease Medication in Nigeria—findings from a descriptive cross-sectional study”

Reviewer 2: Thank you for your comments on this, we asked both the HCWs and patients same sets of questions on the preference of smart lockers in order to get an insight from HCWs who provide services and patients who receive services.

In addition HCW participants were exclusively asked on the suitable personnel for loading the smart lockers. Adjust lastly to reflect changes

Reviewer 2: Thank you for your question, the patients with these chronic diseases were examined. 

728 (Table1)

Reviewer 2: Thank you for your feedback, this adjustment has been made to reflect 5 states as conducted in the study. 

Thank you for pointing this out. “Participants have been included after 1,133 in line 30 in the manuscript”

Reviewer 2: Thank you and this is noted. Absolute numbers have been included in lines 31 and 32 

Reviewer 2: The comment is appreciated, the statement on line 32 has been re-worded as findings.

Reviewer 2: Thank you for your response, it was 1,180 responses. The responses were received through electronic mails, social media platforms such as WhatsApp, LinkedIn, Instagram, and Facebook.

Reviewer 2: Thank you for your feedback on this, Line 91-97 have been removed and the sectionized problem statement and justification have been consolidated into the introduction.

Reviewer 2: Your response on this is deeply appreciated, the objectives statement has been adjusted to reflect just the objective and the numbering has been removed.

Reviewer 2: Thank you for your comments on this. The survey was conducted within the sites of the Implementing partners (IPs) not exclusively to the patients and healthcare workers of the Implementing Partners. Secondly, both the HCWs and patients answered the same sets of questions. Additionally, HCW participants were exclusively asked on the suitable personnel for loading the smart lockers.

Reviewer 2: Thank you for noting this comment, the participants were people above the age of 18 and children were not involved.

Reviewer 2: Thank you for pointing this out the lines 162-164inclusive of the study population have been merged into one statement.

Reviewer 2: Thank you for your comments on this, the survey was conducted within the sites of the Implementing partners (IPs) not exclusively to the patients and healthcare workers of the Implementing Partners. 

Reviewer 2: This comment is appreciated. The ethical statement has been adjusted including adjustment to line 202.

Reviewer 2: Thank you for this comment on the titles of the tables. The titles of the tables have been adjusted as suggested. 

Reviewer 2: Thank you, this is noted. The titles of the tables have been adjusted. The age band in this category was 18-35 years of age not limited to 18 years.

Reviewer 2: The basis for the selection of healthcare workers was the healthcare workforce that manages chronic disease at the clinics.

Reviewer 2: Thank you for your comment on what HCW/patients might benefit from using smart lockers, this has been addressed.

Reviewer 2: Thank you for noting this in this comment, tables 2 and 4 have been consolidated.

Reviewer 2: Thank you for your comments. The difference in lines 280-282 is addressed. 

Reviewer 2: We are appreciative of your comment about the discussion, our discussion section has been adjusted accordingly.

Reviewer 3: Thank you. Further information has been provided in the background concerning traditional method of dispensing drugs.

Reviewer 3: Thank you for your comment, the number of states has been adjusted accordingly.

Reviewer 3: Thank you for your response. The participants were selected or identified through implementing partners in the country. These partners manage healthcare facilities within the 5 States captured in the study.

Reviewer 3: Thank you for your comment. This is noted.

Reviewer 3: Thank you for this comment. This is noted and adjusted.

Reviewer 3: Thank you for your comment on the questionnaire. The questionnaire was self-administered through electronic mail, social media networking sites and groups, through platforms, such as WhatsApp, Facebook, Instagram, and LinkedIn. The smart lockers will be both manually and digitally operated. 

Reviewer 3: Thank you for this comment on sharing the TIFF format for figures within the document, but currently, this is not applicable as there are tables in the document but no figures.

We look forward to hearing from you in due time regarding our submission and to responding to any further questions and comments you may have.

Sincerely,

Ibrahim Bola Gobir.

---

## [Decision Letter · Decision Letter 1]

4 Mar 2024

PONE-D-23-39195R1Patients and Healthcare Workers' Preference for Using Smart Lockers in Dispensing Chronic Disease Medication in Nigeria: Findings from a Descriptive Cross-Sectional StudyPLOS ONE

Dear Dr. Gobir,

Thank you for submitting your manuscript to PLOS ONE. After careful consideration, we feel that it has merit but does not fully meet PLOS ONE’s publication criteria as it currently stands. Therefore, we invite you to submit a revised version of the manuscript that addresses the points raised during the review process.

We look forward to receiving your revised manuscript.

Kind regards,

Mohammed Shuaib

Academic Editor

PLOS ONE

Additional Editor Comments:

This paper talks about an important topic and has considered some of the feedback from Reviewer 2. However, it looks like not all the comments have been dealt with. Please look at Reviewer 2's old and new comments again and update the manuscript .

Reviewers' comments:

Reviewer's Responses to Questions

**Comments to the Author**

1. If the authors have adequately addressed your comments raised in a previous round of review and you feel that this manuscript is now acceptable for publication, you may indicate that here to bypass the “Comments to the Author” section, enter your conflict of interest statement in the “Confidential to Editor” section, and submit your "Accept" recommendation.

Reviewer #2: (No Response)

2. Is the manuscript technically sound, and do the data support the conclusions?

Reviewer #2: Yes

3. Has the statistical analysis been performed appropriately and rigorously? 

Reviewer #2: Yes

4. Have the authors made all data underlying the findings in their manuscript fully available?

Reviewer #2: Yes

5. Is the manuscript presented in an intelligible fashion and written in standard English?

Reviewer #2: No

6. Review Comments to the Author

Reviewer #2: General comments:

Thank you for the opportunity to review the revised manuscript. The revised manuscript, documenting patients’ and health care workers’ preferences in accessing and dispensing chronic disease medications, reads better than the original draft.

The authors have addressed some of the prior comments, but the paper can be further improved by addressing some omissions, removing many distracting subheadings, and restructuring the discussion. Of note, the authors have not addressed key comments and have not explained why. While this paper is improved, the discussion section still needs revision.

As noted in my initial review, this paper addresses a topic of great importance regarding access to HIV and chronic disease medications. This innovation could help resource constrained countries accommodate the growing population of individuals needing medications for chronic diseases. As such, the authors should be given another opportunity to revise the paper

Title:

Line 1: Consider adding the word “accessing” to the title. So, the title can read “Patients and Healthcare Workers' Preference for Using Smart Lockers in Accessing and Dispensing Chronic Disease Medication in Nigeria: Findings from a Descriptive Cross-Sectional study”

Abstract:

Line 30, under results, the word “participants” should be added before the brackets.

Line 47, the sentence is incomplete. The two sentences could be combined or the first one revised to be meaningful.

Background:

Well written and addressed previous comments but can be shortened.

Line 113: The multiple subtitles are distracting. For example, there is no need to put the title “problem statement”. It’s supposed to be part of the introduction and could be removed for better flow.

Line 126: Similarly, the subtitle “justification” is not required. Its usually part of research proposals and not part of manuscripts

Line 140--143: please present as a sentence for easy reading. There is one objective, its not necessary to number one item. This is usually the last part of the background section

Methods

All comments addressed.

Results:

No comments. All prior comments addressed

Discussion:

The discussion should start with why the study was conducted, your key findings and discussion of the findings.

Acknowledgement section:

Written too informally. Could be shortened with a simple sentence that read “The authors would like to thank all those who contributed in one way or another to the collection and analysis of this data

7. PLOS authors have the option to publish the peer review history of their article (what does this mean?). If published, this will include your full peer review and any attached files.

Reviewer #2: No

---

## [Author Response · Author response to Decision Letter 1]

12 Mar 2024

Response to reviewer 2's comments 

Thank you for taking the time to review our manuscript titled “Patients and Healthcare Workers' Preferences for Using Smart Lockers in Accessing and Dispensing Chronic Disease Medication in Nigeria: Findings from a Descriptive Cross-Sectional Study”, submitted to PLOS ONE We appreciate your insightful comments and constructive feedback, which have greatly contributed to improving the quality of our work.

Reviewer 2: Thank you for this response, The title has been revised as suggested.

Reviewer 2: Thank you for your comment. The word “participants” has been included in the bracket.

Reviewer 2: We appreciate your feedback. The statement has been adjusted to sound more meaningful.

Reviewer 2: Your comment is appreciated, in the background of our study, the paragraphs have been adjusted. 

Reviewer 2: Thank you for your comment. This adjustment was made during the previous revision and the subtitles merged into the background.

Reviewer 2: We are appreciative of your feedback. This adjustment was made during the previous revision, and the sections were merged into the background.

Reviewer 2: Thank you for your comment. This adjustment to the methods was made during the previous revision.

Reviewer 2: Thank you for your response. The discussion session has been adjusted.

Reviewer 2: Thank you for your suggestion on the acknowledgement. It has been adopted.

---

## [Decision Letter · Decision Letter 2]

5 Apr 2024

PONE-D-23-39195R2Patients and Healthcare Workers' Preferences for Using Smart Lockers in Accessing and Dispensing Chronic Disease Medication in Nigeria: Findings from a Descriptive Cross-Sectional StudyPLOS ONE

Dear Dr. Gobir,

Thank you for submitting your manuscript to PLOS ONE. After careful consideration, we feel that it has merit but does not fully meet PLOS ONE’s publication criteria as it currently stands. Therefore, we invite you to submit a revised version of the manuscript that addresses the points raised during the review process.

The paper addresses a significant topic concerning access to HIV and chronic disease medications, which underscores its importance in healthcare. However, it's concerning that some comments from the reviewer have not been adequately addressed in the latest round of revisions. It's crucial for the authors to take seriously the comments provided by the reviewer in previous rounds

Table 1, 2, and 3 require graphical representation to enhance the presentation of results and facilitate easier comprehension for readers. Incorporating visual aids such as graphs or charts can help convey complex information more effectively and improve the overall readability of the manuscript.

Additionally, while the paper is generally well-written, there may be some small errors that need to be addressed during proofreading. It's essential to ensure that the manuscript is free from grammatical errors, typos, and formatting issues before final submission.

We look forward to receiving your revised manuscript.

Kind regards,

Mohammed Shuaib

Academic Editor

PLOS ONE

**Additional Editor Comments:**

Background section:

1. Streamline the background section by eliminating unnecessary details and redundant phrases.

2. Break down lengthy sentences into shorter, more digestible segments to enhance readability.

3. Focus on highlighting the gap in existing literature regarding the use of smart lockers in Nigeria and emphasize the potential benefits of implementing such systems in healthcare.

Study Design section:

1. Provide additional context to explain why the descriptive cross-sectional design was chosen and how it aligns with the broader study objectives.

2. Clarify by adding additional paragraph the criteria used for selecting healthcare workers and patients to ensure consistency with the study's objectives.

3. Justify by adding additional paragraph the method used for determining the sample size, addressing any limitations associated with proxy estimates and assumptions.

Data Analysis section:

1. Offer more detail by adding additional paragraph on the specific variables analyzed during the study and the statistical tests performed.

2. Describe any assumptions made during the data analysis process and explain the rationale behind selecting Stata version 15 for analysis.

Discussion section :

1. Emphasize the study's objective of addressing the lack of research on smart locker implementation for chronic disease medication in Nigeria, highlighting its potential to alleviate healthcare burdens.

2. Discuss the overwhelming preference for smart locker use among persons living with HIV aged 18 to 40, underscoring the need for decentralized care models and the familiarity with community-based drug delivery.

3. Address the practicality by adding additional paragraph of one-on-one counseling for patient orientation on smart locker use, considering its effectiveness in overcoming barriers to understanding and the importance of ensuring patient adherence to medication regimens.

Reviewers' comments:

Reviewer's Responses to Questions

**Comments to the Author**

1. If the authors have adequately addressed your comments raised in a previous round of review and you feel that this manuscript is now acceptable for publication, you may indicate that here to bypass the “Comments to the Author” section, enter your conflict of interest statement in the “Confidential to Editor” section, and submit your "Accept" recommendation.

Reviewer #2: All comments have been addressed

2. Is the manuscript technically sound, and do the data support the conclusions?

Reviewer #2: Yes

3. Has the statistical analysis been performed appropriately and rigorously? 

Reviewer #2: I Don't Know

4. Have the authors made all data underlying the findings in their manuscript fully available?

Reviewer #2: Yes

5. Is the manuscript presented in an intelligible fashion and written in standard English?

Reviewer #2: Yes

6. Review Comments to the Author

Reviewer #2: (No Response)

7. PLOS authors have the option to publish the peer review history of their article (what does this mean?). If published, this will include your full peer review and any attached files.

Reviewer #2: No

---

## [Author Response · Author response to Decision Letter 2]

24 Apr 2024

Thank you for providing valuable feedback and allowing us to resubmit a revised manuscript draft titled "Patients and Healthcare Workers' Preferences for Using Smart Lockers in Accessing and Dispensing Chronic Disease Medication in Nigeria: Findings from a Descriptive Cross-Sectional Study." Your thorough review has undoubtedly enhanced the quality and clarity of our work. A graphical representation has been provided to enhance the presentation of results for Tables 1, 2, and 3. The paper has been proofread and readjusted to remove errors.

Below, we have provided a detailed response to each point raised, outlining the revisions made to strengthen the manuscript.

Thank you for your comment. A graphical representation has been provided to enhance the presentation of results for Tables 1, 2, and 3

Thank you for your comment. The paper has been proofread and readjusted to remove errors.

Thank you for your suggestion. No changes were made to the financial disclosure statement.

Your comment on the background is well appreciated. The background of the study has been streamlined as requested.

Your comment is well received. Lengthy sentences have been made shorter and concise as prescribed.

Thank you for your comment, existing gaps in literature regarding our study is captured in the latter of the background section on page 5

Thank you for your comment on the study design. Additional context to explain why the descriptive cross-sectional design has been captured in the method section on page 6, this is captured through an explanation of our study population.

Your comment is very well appreciated on the inclusion criteria. Additional context has been provided in the inclusion criteria captured on page 6

Thank you for your comment. An additional paragraph has been added in the method section describing the selection of participants. The sample size was for both healthcare workers and patients have been included in the section as well.

Your comment is very well appreciated and received. An additional statement has been provided under the paragraph for each variable analyzed during the study and the statistical test performed on page 7.

Thank you for your comment. We have explained in the methods section on page 8, the assumptions made during the data analysis and the justification for using Stata 15 for analysis.

Your comment is well received and appreciated. On page 15 emphasis on the lack of research on smart lockers and implementation has been included as a preamble to discussing our findings.

Thank you for your comment. The preference for smart lockers among persons living with HIV aged 18-40 among most suitable settings is included in line 261 – 268

Thank you for your observation. The practicality of one-on-one counselling for patient orientation is discussed on page 17 lines 291-309

---

## [Decision Letter · Decision Letter 3]

30 Apr 2024

Patients and Healthcare Workers' Preferences for Using Smart Lockers in Accessing and Dispensing Chronic Disease Medication in Nigeria: Findings from a Descriptive Cross-Sectional Study

PONE-D-23-39195R3

Dear Dr. Gobir,

We’re pleased to inform you that your manuscript has been judged scientifically suitable for publication and will be formally accepted for publication once it meets all outstanding technical requirements.

Kind regards,

Mohammed Shuaib

Academic Editor

PLOS ONE

Additional Editor Comments (optional):

Reviewers' comments:

Reviewer's Responses to Questions

**Comments to the Author**

1. If the authors have adequately addressed your comments raised in a previous round of review and you feel that this manuscript is now acceptable for publication, you may indicate that here to bypass the “Comments to the Author” section, enter your conflict of interest statement in the “Confidential to Editor” section, and submit your "Accept" recommendation.

Reviewer #2: All comments have been addressed

2. Is the manuscript technically sound, and do the data support the conclusions?

Reviewer #2: Yes

3. Has the statistical analysis been performed appropriately and rigorously? 

Reviewer #2: I Don't Know

4. Have the authors made all data underlying the findings in their manuscript fully available?

Reviewer #2: Yes

5. Is the manuscript presented in an intelligible fashion and written in standard English?

Reviewer #2: Yes

6. Review Comments to the Author

Reviewer #2: (No Response)

7. PLOS authors have the option to publish the peer review history of their article (what does this mean?). If published, this will include your full peer review and any attached files.

Reviewer #2: No

---

## [Editor Report · Acceptance letter]

27 Jun 2024

PONE-D-23-39195R3 

PLOS ONE

Dear Dr. Gobir, 

I'm pleased to inform you that your manuscript has been deemed suitable for publication in PLOS ONE. Congratulations! Your manuscript is now being handed over to our production team.

Kind regards, 

on behalf of

Dr. Mohammed Shuaib 

%CORR_ED_EDITOR_ROLE%

PLOS ONE